# The Relationship between Career Decision-Making Self-Efficacy, Career Preparation Behaviour and Career Decision Difficulties among South Korean College Students

Sanghee Lee [1], Jaeeun Jung [2], Sungeun Baek [3] and Songyi Lee [1,*]

1   Department of Counseling and Coaching, Graduate School, Dongguk University, Seoul 04620, Korea
2   Department of Early Childhood Education, Dongyang University, Yeongju-si 36040, Korea
3   Department of Police and Criminal Psychology, Dongyang University, Yeongju-si 36040, Korea
*   Correspondence: songyilee@empas.com or songyilee@dongguk.edu; Tel.: +82-10-6357-7310

**Abstract:** Few studies have continuously examined the relationship between career decision-making self-efficacy variables and career-related variables in South Korea's specific cultural context. Accordingly, this study aims to analyse (using Pearson's correlations and structural equation modelling) the relationships between South Korean college students' career decision-making self-efficacy, career preparation behaviour, and career decision difficulties. There were positive and negative relationships between career decision-making self-efficacy and career preparation behaviour career decision difficulties, respectively. In addition, we found a positive effect between career preparation behaviour and career decision-making self-efficacy, while career decision difficulties negatively affected career decision-making self-efficacy. Considering the standardised coefficient of the specific direct effect, the effect on career decision-making self-efficacy of career preparation behaviour was larger than that of career decision difficulties. It is recommended that career programmes are developed that help college students to independently set their career goals, actively search for career information, and promote career preparation behaviour while considering their majors. It is also recommended career counselling programmes be designed that can help them establish their self-concept and identity. These findings could provide the necessary basic data for the construction of an effective college career guidance system and inform strategies for improving college students' career decision-making self-efficacy.

**Keywords:** career decision-making self-efficacy; career preparation behaviour; career decision difficulties; college students; South Korea

## 1. Introduction

The economic crisis caused by the COVID-19 pandemic has affected unemployment [1], which has created a challenging situation for college students who are preparing for employment, necessitating increased time and resource costs during job hunting. Therefore, effective career preparation behaviour for appropriate career decisions has become an important issue for college students [2]. Studies have been conducted on how college students can make effective career decisions in a changing society [3–5].

In South Korea, college education has become commonplace, and in 2021, 73.7% of high-school graduates entered college [6]. However, due to the excessive studies required for entrance exams, most high-school students enter college without having considered their aptitudes for their future career trajectory. Instead, they postpone career decisions until when they are already in college [7]. Consequently, college students often lose interest in their studies, change their major, or even entirely withdraw from college. They may struggle with 'spec-building' activities in their search for a stable or preferential workplace, and experience difficulties with career preparation [8–11]. According to Sohn [12], a higher proportion of Korean college students, when compared to Chinese or American students,

were unable to concentrate on the career preparation process because they could not choose a career path. They competitively prepare for entrance exams with the goal of attending an 'excellent university', following degreeocracy and arrivism tendencies. In addition, in a society with high employment uncertainty, they are preparing competitively for 'high-quality jobs'. This phenomenon of 'career funnelling' is caused by various factors, such as unstable employment environment, educational inflation, mismatch of manpower, and major by field, as well as a distorted view that disregards specific occupations [13]. Therefore, institutional and policy efforts are required to help colleges and the state properly prepare college students for their sustainable career development.

Significant career-related variables include career decision-making self-efficacy, career barriers, career preparation behaviour, career attitude maturity, and achievement motivation [14,15]. Among these variables, career decision-making self-efficacy significantly influences the understanding of career development stages. Young people are highly interested in career choices, especially those they consider congruent with their choice of major [16,17]. Career decision-making self-efficacy extends from self-efficacy to confidence by successfully performing tasks related to general career decisions [18]. Hackett and Betz applied self-efficacy to careers based on Bandura's social cognitive theory [16], proposing 'career self-efficacy' as a new construct [18]. Self-efficacy comprises the belief in one's own ability together with concrete and practical skills. When encountering obstacles, individuals decide whether to act, how much effort to expend, and how much to endure based on their expectations [19]. Hackett and Betz further applied self-efficacy theory to career counselling to develop a scale for professional self-efficacy [18], from which Taylor and Betz introduced the concept of 'career decision-making self-efficacy' [20]. This form of self-efficacy refers to rational decision-making that involves the self-conviction necessary to make appropriate decisions in inevitable and ambiguous situations [21].

Career decision-making self-efficacy plays a mediatory role in young people's career concerns and career commitment. It is also an important predictor for career exploration [22,23]. Career decision-making self-efficacy has an impact on resolving difficulties faced at work and improving perseverance while remaining in work [24]. Thus, self-efficacy plays a crucial role during undergraduate study as students undergo the career development process [18].

Speas defined career preparation behaviour as the process by which career cognition transforms into practical action [25]. While this process was regarded as a practical action that individuals should perform to make the right career decisions [26], such a definition implies the variable nature of behaviour that emphasises the cognitive and behavioural aspects of career decision-making. Therefore, career preparation behaviour is an important variable for college students tasked with making career decisions to obtain appropriate employment [27–30]. College students use the internet, advice from friends and family, and courses that provide career information when preparing for their career [31]. Considering the information collection behaviour, of which carrier preparation behaviour is a sub-factor, the action of seeking actual resources is also important for college students' career preparation.

Moreover, career preparation behaviour among college students has a strong relationship with career decision-making self-efficacy [32,33]. However, South Korean college students generally exhibit infrequent career preparation behaviours [34,35], which means that South Korean college students experience difficulties when actively exploring, setting goals, and executing actions to prepare for their chosen careers [36,37]. Korea's college entrance exam-oriented guidance hinders Korean students from establishing career identities and causes difficulties in career preparation behaviour. Career preparation behaviour refers to various foundational activities that individuals perform to either set or achieve personal career goals [26,38,39]. Although this behaviour is a lifelong developmental task, it is of particular importance for college students. Career preparation is a time to make career goals concrete, converting tentative job preferences into concrete ones by completing related education and training, and executing their transition to the labour market [40]. This can also play a crucial role in building confidence both in their choice of major and in

job selection, facilitating the efficient achievement of selected career goals [41]. Preparations include cognitive and attitudinal aspects, and actual behavioural aspects [26]. Thus, for behavioural changes to concretely and practically occur, career preparation behaviour is a necessary cognitive and practical precursor of career decision-making self-efficacy.

Deciding on a career path and preparing for the future is one of the most important developmental tasks for college students. However, the modern job market is rapidly changing due to the development of cutting-edge technologies, such as AI and big data. These changes are aggravating emotional difficulties in career decision-making. The term 'career decision-making difficulties' refers to the general difficulties an individual experiences in career-related decision-making [42].

The emotional- and personality-related career decision-making difficulties (EPCD) scale refers to difficulties that prevent individuals from making career decisions related to career counselling [42]. The EPCD scale measures aspects such as personality difficulties and psychological problems, including pessimistic perspectives, identity, self-concept, and anxiety. Kulcsár et al. found an association between career decision-making difficulties and negative emotions among adolescents [43]. However, decision-making can be rendered more effective when psychological problems are resolved [44]. In other words, emotions and self-confidence play a vital role in career decision-making, and any associated difficulties have a strong relationship with career decision-making self-efficacy. A higher level of EPCD is associated with a lower level of career decision-making self-efficacy [45]. Career decision self-efficacy primarily mediates the relationship between overall and specific abilities of emotional intelligence and career decision-making difficulties [46]. For South Korean college students, these difficulties seem to be due to their devotion to preparing for college entrance exams rather than forming their career identity during adolescence [47]. Therefore, it is necessary to identify relevant variables to understand the difficulties faced by college students in career decision-making and examine the causes of those difficulties.

Although career decision-making is a vital developmental task for college students, South Korea's social environment is not conducive to career exploration and rarely provides opportunities for effective career support planning [48]. Therefore, colleges should provide a foundation for independence in adulthood, supporting college students based on fostering their career decision-making self-efficacy. Most studies conducted in South Korea set career decision-making self-efficacy as an independent or mediating variable, and set career-related variables, such as career preparation behaviour and career decision difficulties, as dependent variables. In the cultural context of South Korea, no studies have attempted to continuously determine the relationship between variables. Cultural values can be an important factor in career decision-making [49]. In particular, Korean adolescents often lack self-understanding and have difficulties in deciding their career path because they receive guidance for college entrance exams rather than receiving appropriate career guidance and counselling based on their aptitudes and interests [50]. Therefore, this study aims to confirm the relationships between identified factors affecting career decision-making self-efficacy among college students. This will provide the necessary basic data for the construction of an effective college career guidance system. Accordingly, we aim to address the following research questions:

**Research Question 1:** What is the relationship between career decision-making self-efficacy, career preparation behaviour, and career decision difficulties among South Korean college students?

**Research Question 2:** What are the relative effects of career preparation behaviour and career decision difficulties on career decision-making self-efficacy among South Korean college students?



## 2. Methods

### 2.1. Participants

The study targeted South Korean college students. Only college students who were readily available, accessible, and willing to be part of the study were recruited in Seoul and Gyeonggi-do. The sample comprised 341 South Korean college students randomly selected from the 2020 academic year, of whom 169 were female (49.6%) and 172 were male (50.4%). The students' ages were in the range of 19–24 years (mean age 21.1; SD = 1.7), comprising the following groups: 19 years (n = 89; 26.1%), 20 years (n = 70; 20.5%), 21 years (n = 40; 11.7%), 22 years (n = 51; 15.0%), and 23 years (n = 51; 15%). There were 132 (38.7%) freshmen, 122 (35.7%) sophomores, 50 (14.7%) juniors, and 37 (10.9%) seniors. In addition, 64 (18.8%) majored in the humanities, 78 (22.9%) majored in the social sciences, 37 (10.9%) majored in the natural sciences, 121 (35.5%) majored in engineering, and 41 (12.0%) majored in other areas.

### 2.2. Measurement

We used the career decision-making self-efficacy scale (CDMSES), the career preparation behaviour scale, the EPCD scale, and a personal information form for data collection.

#### 2.2.1. Career Decision-Making Self-Efficacy

The CDMSES [51] was developed as a Korean version of the original scale developed by Taylor and Betz [52]. The Korean version of CDMSE-SF has been frequently used in various related studies in Korea [53–55]. Using Likert-type responses, it measures individuals' confidence in their ability to make career decisions and consists of five sub-dimensions: job information collection, goal selection, plan establishment, problem-solving, and self-evaluation. Each of the five sub-dimensions comprises five items (25 items in total), all measured on a 5-point Likert scale. Table 1 presents the concept and reliability of the sub-dimensions.

**Table 1.** Sub-dimensions of the Career Decision-making Self-efficacy Scale and reliability.

| Sub-Dimension | Concept | Reliability |
|---|---|---|
| 1. Job information collection | Respondents' confidence that they can find their preferred jobs and concretely explore the elements of the relevant jobs | 0.746 |
| 2. Goal selection | Respondents' confidence in choosing their study and career paths | 0.835 |
| 3. Plan establishment | Respondents' belief that they can independently establish and carry out plans related to school admission and job searching | 0.823 |
| 4. Problem-solving | Respondents' belief that they can independently resolve difficult situations | 0.799 |
| 5. Self-evaluation | Respondents' confidence in evaluating their abilities and needs and deciding whether a job suits them | 0.747 |
| | Total | 0.948 |

#### 2.2.2. Career Preparation Behaviour

Career preparation behaviour involves concrete and practical actions and cognitions related to individuals' optimal career decision-making and achievement of personal career goals following career decisions [25]. Based on the Career Exploration Survey (CES) [56], the Vocational Questionnaire II [57], and the Career Planning Questionnaire [58], Kim developed the Career Preparation Behaviour Scale [52] in South Korea by analysing the behaviours of college students who decided on their careers after career counselling. A validation study of the Career Preparation Behaviour scale was also conducted by Kim, B.H. [59], and this scale has been frequently used in various related studies in Korea [60,61]. The scale comprises three sub-dimensions: information-collection behaviour (six items), tool-preparation behaviour (five items), and practical efforts to achieve career goals (seven items)—18 items in total—

all measured on a 5-point Likert scale. Table 2 presents the content and reliability of the scale's sub-dimensions.

**Table 2.** Sub-dimensions of the Career Preparation Behaviour Scale and reliability.

| Sub-Dimension | Concept | Reliability |
| --- | --- | --- |
| 1. Information-collection behaviour | Efficiently collecting information on one's abilities, aptitudes, interests, personality, etc.; information about the current situation; prospects of the occupational group in which one is interested | 0.832 |
| 2. Tool-preparation behaviour | Purchasing the textbooks, equipment, etc., and attaining the necessary qualifications and licenses for their preferred job | 0.769 |
| 3. Practical efforts to achieve career goals | Investing time and effort concretely and practically to attain a target job | 0.849 |
| | Total | 0.942 |

### 2.2.3. Career Decision Difficulties

The Korean short version of the EPCD-28 scale (K-EPCD-28) [62] is a Likert-type scale developed by reducing the EPCD scale [42]; it comprises three sub-dimensions: pessimistic perspective (eight items), anxiety (10 items), and self-concept and identity (10 items)—28 in total—all measured on a 9-point Likert scale. K-EPCD-28 was developed and validated by Min and Kim [63]. It has been frequently used in various related studies in Korea [62,64]. Table 3 presents the concepts and reliability of the K-EPCD-28 sub-dimensions.

**Table 3.** Sub-dimensions of the Career Decision Difficulties Scale and reliability.

| Sub-Dimension | Concept | Reliability |
| --- | --- | --- |
| 1. Pessimistic perspective | Pessimistic perspectives on the occupational world and individuals' control | 0.834 |
| 2. Anxiety | Anxiety about outcomes, such as selection processes and uncertainties | 0.937 |
| 3. Self-concept and identity | Trait anxiety, self-esteem, undifferentiated identity, conflicting attachment, and separation, etc. | 0.913 |
| | Total | 0.953 |

### 2.3. Personal Information Form

We prepared a personal information form consisting of five structured questions to establish the students' age, gender, type of school, grade, and major. In line with the principle of confidentiality, we did not reveal any identity information and subjects were indicated by numbers in the study results.

### 2.4. Procedure

We created an online questionnaire containing the described scales, including purpose of the research, personal information form, and informed consent form. The researchers posted a link to the created online questionnaire on the notice board of the e-class based on a learning management system (LMS) in the research target school. In the study, the informed consent form was also included in some of the questionnaires. The present study was conducted in compliance with the Committee on Publication Ethics (COPE), as well as the guidelines of the Declaration of Helsinki. Signed informed consent forms were obtained from our participants after a thorough explanation of the study's aims and procedures. A total of 658 students from two colleges in Seoul and Gyeonggi-do, through convenience sampling, could access the LMS for the study. During the week of December 2020, 350 students responded to the online questionnaire. An amount of 9 responses with unanswered items and/or errors were removed, leaving a total of 341 responses for analysis.

*2.5. Data Analysis*

We assessed the reliability of the measurement tool scores by examining Cronbach's α coefficient using Statistical Package for the Social Sciences (SPSS) 21.0 (IBM, New York, NY, USA); we used 0.70 as the cut-off for acceptable reliability [65]. Descriptive statistics regarding the mean (M), standard deviation (SD), skewness, and kurtosis were analysed. Furthermore, we conducted a Pearson correlation coefficient analysis.

We conducted first-order confirmatory factor analysis (CFA) with Mplus 7.2 (Muthén & Muthén, Los Angeles, CA, USA) to examine how well the independent and dependent dimensions with their related reflective sub-dimensions fit the present data. In addition, we conducted structural equation modelling (SEM) using Mplus 7.2. We examined whether all the latent variables (dimensions) were significantly related in theoretically consistent ways. We hypothesised that career preparation behaviour and career decision difficulties (i.e., the predictors in the model) would exert influence on career decision-making self-efficacy.

We examined the fit of these structural models via the root mean square error of approximation (RMSEA) [66]; standardised root mean squared residual (SRMR) [67]; comparative fit index (CFI) [68]; and the Tucker–Lewis index (TLI) [69]. CFI and TLI values greater than 0.90 [68] and CFI and TLI values greater than 0.95 [70] were used as benchmarks for acceptable and good model fit, respectively. RMSEA and SRMR values lower than 0.08 and lower than 0.05 were used as benchmarks for acceptable and good fit, respectively [70,71]. A standardised coefficient was used to estimate the difference in specific direct effects [72].

## 3. Results

*3.1. Preliminary and Descriptive Analysis*

Prior to conducting the main analysis, we ran a preliminary analysis in which we examined whether the participants' characteristics (i.e., gender, grade, and major) affected the homogeneity of the analysed sample. Specifically, we conducted Spearman correlation analysis to confirm whether participants' characteristics were related to the variables of interest. Gender was related to anxiety, and grade was related to self-evaluation and tool preparation behaviour. Major was related to information collection behaviour, to practical efforts to achieve career goals, and to anxiety. However, it can be concluded that the correlation coefficient was ≤0.30, indicating only a slight—if any—relationship [73]. In conclusion, the slight—if any—relationship from Spearman correlation analysis confirmed that the unequal participants' characteristics did not present a serious problem in our subsequent analysis.

We assessed the descriptive statistics. Table 4 represents respondents' mean (M) and standard deviation (SD) values. Table 4 also includes the skewness and kurtosis of the data distribution where no values exceeded the cut-off scores [74], assuring the variables' normal distribution.

Table 4 also shows the results of Pearson's analysis to determine the relationship between variables of interest. Career decision-making self-efficacy was significantly positively correlated with career preparation behaviour ($r = 0.709$, $p < 0.01$) and its sub-dimensions: information collection behaviour ($r = 0.695$, $p < 0.01$), tool preparation behaviour ($r = 0.647$, $p < 0.01$), and practical efforts to achieve career goals ($r = 0.628$, $p < 0.01$). In contrast, career decision-making self-efficacy was significantly negatively correlated with career decision difficulties ($r = -0.230$, $p < 0.01$) and its sub-dimensions: pessimistic perspective ($r = -0.128$, $p < 0.05$), anxiety ($r = -0.222$, $p < 0.05$), and self-concept and identity struggles ($r = -0.238$, $p < 0.01$).

**Table 4.** Correlations among measures and descriptive statistics.

| Dimensions | 1 | 2 | 3 | 4 | 5 | 6 | 7 | 8 | 9 | 10 | 11 | 12 | 13 | 14 |
|---|---|---|---|---|---|---|---|---|---|---|---|---|---|---|
| career preparation behaviour | 1 | | | | | | | | | | | | | |
| information collection behaviour | 0.897 *** | 1 | | | | | | | | | | | | |
| tool preparation behaviour | 0.927 *** | 0.785 *** | 1 | | | | | | | | | | | |
| practical efforts to achieve career goals | 0.937 *** | 0.730 *** | 0.806 *** | 1 | | | | | | | | | | |
| career decision difficulties | −0.139 * | −0.162 ** | −0.134 * | −0.098 | 1 | | | | | | | | | |
| pessimistic perspective | −0.039 | −0.096 | −0.053 | 0.020 | 0.813 *** | 1 | | | | | | | | |
| anxiety | −0.150 ** | −0.141 ** | −0.131 | −0.142 ** | 0.913 *** | 0.615 *** | 1 | | | | | | | |
| self-concept and identity | −0.152 ** | −0.179 *** | −0.152 ** | −0.103 | 0.916 *** | 0.647 *** | 0.748 *** | 1 | | | | | | |
| career decision-making self-efficacy | 0.709 *** | 0.695 *** | 0.647 *** | 0.628 *** | −0.230 *** | −0.128 | −0.222 *** | −0.238 *** | 1 | | | | | |
| job information collection | 0.653 *** | 0.634 *** | 0.582 *** | 0.593 *** | −0.193 *** | −0.124 | −0.165 *** | −0.210 *** | 0.911 *** | 1 | | | | |
| goal selection | 0.652 *** | 0.623 *** | 0.590 *** | 0.592 *** | −0.254 *** | −0.151 *** | −0.269 *** | −0.230 *** | 0.899 *** | 0.782 *** | 1 | | | |
| future plan establishment | 0.674 *** | 0.658 *** | 0.625 *** | 0.592 *** | −0.228 *** | −0.114 | −0.224 *** | −0.240 *** | 0.921 *** | 0.832 *** | 0.778 *** | 1 | | |
| problem solving | 0.608 *** | 0.607 *** | 0.567 *** | 0.523 *** | −0.139 | −0.066 | −0.122 | −0.164 *** | 0.878 *** | 0.731 *** | 0.687 *** | 0.760 *** | 1 | |
| self-evaluation | 0.608 *** | 0.611 *** | 0.550 *** | 0.530 *** | −0.226 *** | −0.125 | −0.224 *** | −0.230 *** | 0.907 *** | 0.772 *** | 0.806 *** | 0.773 *** | 0.781 *** | 1 |
| M | 59.03 | 21.28 | 16.37 | 21.38 | 139.25 | 37.87 | 52.26 | 49.12 | 92.84 | 18.54 | 18.75 | 18.11 | 18.41 | 19.03 |
| SD | 15.235 | 4.996 | 4.662 | 6.869 | 39.125 | 10.976 | 16.950 | 16.073 | 15.288 | 3.307 | 3.624 | 3.639 | 3.539 | 2.819 |
| Skewness | −0.057 | −0.488 | −0.158 | 0.151 | −0.178 | 0.014 | −0.390 | −0.186 | −0.155 | −0.094 | −0.272 | −0.263 | −0.070 | −0.469 |
| Kurtosis | 0.165 | 0.384 | −0.144 | −0.424 | 0.344 | 0.637 | −0.030 | 0.035 | 0.830 | 0.324 | 0.400 | 0.604 | −0.313 | 1.181 |

'*' for $p < 0.05$, '**' for $p < 0.01$, '***' for $p < 0.001$.

### 3.2. Measurement Model

This study adopted a two-step approach to SEM to test the research hypotheses [75]. First, a confirmatory factor analysis (CFA) was conducted to report the statistics for the measurement model. Various goodness-of-fit criteria, as mentioned above, were employed to assess the model's fit to the data. The goodness of fit results demonstrated that the measurement model was a good fit for the data: $\chi2$ (41, N = 341) = 329.810, $p < 0.001$, (SRMR = 0.033, RMSEA = 0.080 (90% CI [0.064, 0.095]), CFI = 0.971 and TLI = 0.961). Figure 1 shows that the factor loadings of all latent variable indicators were significant.

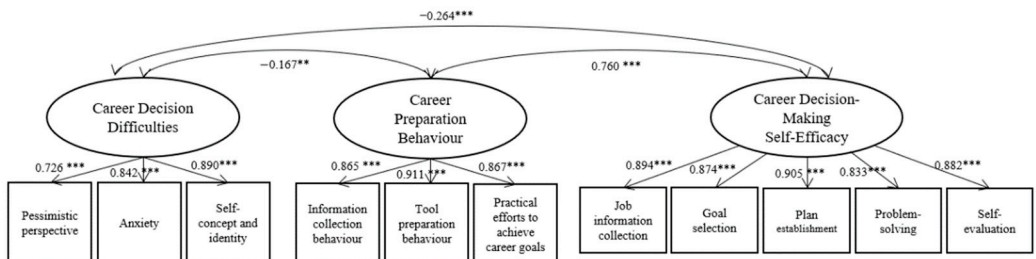

**Figure 1.** Measurement model. '**' for $p < 0.01$, '***' for $p < 0.001$.

The reliability was assessed using Cronbach's α values (as previously mentioned) and composite reliability (CR). All CR values exceeded the recommended cut-off point of 0.7 for all four dimensions–career preparation behaviour (0.912), career decision difficulties (0.861), and career decision-making self-efficacy (0.944), indicating that the data were internally consistent [76]. Moreover, convergent and discriminant validity was successfully obtained from the result. The measuring convergent validity was successfully obtained for all indicators specified to measure a common factor that had relatively high standardised factor loadings (FL) on that factor [77], as shown in Figure 1. Figure 1 shows that all FL values were 0.73–0.91, exceeding the proposed cut-off value of 0.70 [77]. All average variance extracted (AVE) values, career preparation behaviour (0.777), career decision difficulties (0.676), and career decision-making self-efficacy (0.771) exceeded the value of 0.50, also showing satisfactory convergent validity [76]. The discriminant validity was obtained through estimated correlations between factors [77] that were not excessively high (i.e., < 0.90 in absolute value), as shown in Figure 1.

### 3.3. Structural Equation Modelling (SEM)

We used SEM to examine the influences of career preparation behaviour and career decision difficulties on career decision-making self-efficacy. The structural model fit the data well: $\chi2$ (41, N = 341) = 329.810, $p < 0.001$, (SRMR = 0.033, RMSEA = 0.080 (90% CI [0.064, 0.095]), CFI = 0.971, and TLI = 0.961). As shown in Table 5 and Figure 2, the direct effect from career preparation behaviour to career decision-making self-efficacy was significant ($t$-value = 24.867, $p < 0.001$), with a standardised path coefficient of 0.736. The effect of career decision difficulties on career decision-making self-efficacy was significant ($t$-value = −3.344, $p < 0.001$), with a standardised path coefficient of −0.141. The difference between these two direct effects was examined using a standardised path coefficient; the effect of career preparation behaviour on career decision-making self-efficacy was significantly greater than the effect of career decision difficulties on career decision-making self-efficacy. Figure 2 also shows that the explanatory power ($R^2$) of all paths explains 59.7% of the variance in career decision-making self-efficacy.

**Table 5.** Results of the structural model.

| Direct Effect | B | SE | C.R. (*t*-Value) |
|---|---|---|---|
| career preparation behaviour → career decision-making self-efficacy | 0.504 | 0.034 | 14.837 *** |
| career decision difficulties → career decision-making self-efficacy | −0.052 | 0.016 | −3.280 *** |

Model fit: ($\chi$2 (41, N = 447) = 129.810, *p* < 0.001, RMSEA = 0.080 (90% CI [0.064, 0.095]), SRMR = 0.033, CFI = 0.971, TLI = 0.961). B = Non Standard regression, SE = Standard Error, C.R. = Critical Ratio, '***' for *p* < 0.001.

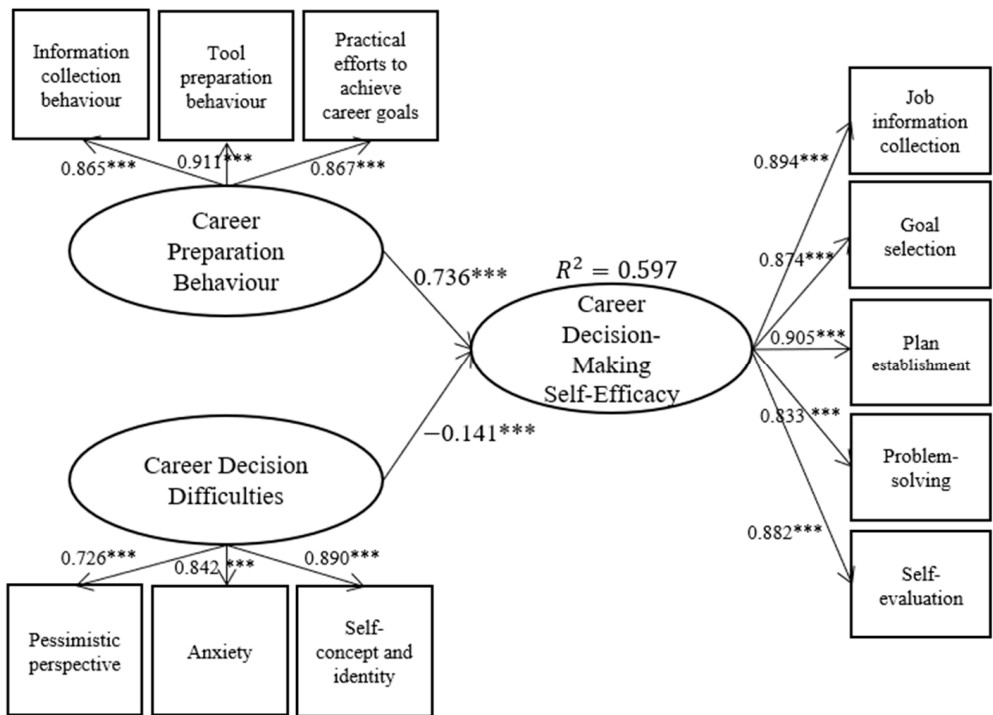

**Figure 2.** The structural model. '***' for *p* < 0.001.

## 4. Discussion

This study investigated the predictive variables and relationship between career preparation behaviour and career decision difficulties. These are particularly important in college life, as they affect career decision self-efficiency, which is reportedly an important factor in career development. Through the results of this study, it is possible to understand the difficulties faced by college students in career decision-making and to identify the causes of these problems. It can also be used as meaningful data when trying to provide appropriate counselling or programmes to help college students with low self-efficacy in career decision-making.

Our first major finding was that there were significant correlations between career decision-making self-efficacy, career preparation behaviour, and career decision difficulties. Second, a positive relationship emerged between career preparation behaviour and career decision-making self-efficacy (i.e., higher levels of career preparation behaviour were related to higher levels of career decision-making self-efficacy). In addition, career decision difficulties negatively predicted career decision-making self-efficacy. Considering the standardised coefficient, career preparation behaviour exhibited the strongest effect on career decision-making self-efficacy, followed by career decision difficulties. Career decision-making self-efficacy and career preparation behaviour are generally recognised as the most important variables related to college students' careers, and numerous studies have reported their positive relationship. Our findings align with previous research, indicating that when students have high career decision-making self-efficacy, they more frequently engage in career preparation behaviour [28–30,78–85]. This shows that career decision-

making self-efficacy increases when students are ready to seek employment. In addition, Park et al. [33] showed that career decision-making self-efficacy had the largest effect on college students' career preparation behaviours; as did Lee [32], in a meta-analysis of career preparation behaviour-related variables. Our findings mean that self-efficacy explains the greatest variance in career preparation behaviour, entailing that career decision-making self-efficacy directly affects career preparation behaviour. Therefore, it is necessary to provide systematic career counselling and programmes that strengthen career preparation behaviour in universities to provide appropriate help for career preparation. In addition, since students' career paths can continuously change, it is necessary to develop practical and specific career counselling and guidance programmes that can increase the self-efficacy of career decision-making, provide various opportunities, and deliver practical systematic career guidance.

The results of this study show that career preparation behaviour exerts the strongest influence on career decision-making self-efficacy. This study defines career preparation behaviour as information-collection behaviour, tool-preparation behaviour, and practical efforts to achieve career goals. Therefore, it is suggested that these three be considered in detail when preparing specific counselling and guidance programmes that can promote career decision-making self-efficacy.

Our research also substantiates the association between higher career decision-making self-efficacy and more frequent information-seeking on career choices, as determined by Blustein [85]. Similarly, this accords with a 2021 study that found that higher career decision-making self-efficacy is positively associated with career exploration behaviour [23]. These findings suggest that college students with stronger beliefs and confidence in their ability to make career decisions engage in either career exploration or preparation behaviour more frequently and experience fewer career decision-making difficulties. Therefore, concerning career preparation behaviour and career decision difficulties, career decision-making self-efficacy is a critical variable. Accordingly, colleges need to construct effective career guidance systems to foster students' career decision-making self-efficacy.

Second, this study demonstrates that information collection behaviour, practical efforts to achieve career goals, and tool preparation behaviour (sub-factors of career preparation behaviour) positively predict career decision-making self-efficacy. Moreover, struggles with self-concept and identity (sub-factors of career decision difficulties) negatively predict it. This discovery means that students experience greater career decision-making self-efficacy if they have greater information collection behaviour, greater practical efforts to achieve career goals, and fewer self-concept and identity struggles. Thus, among the sub-factors of career preparation behaviour, both information collection behaviour and practical efforts to achieve career goals demonstrate explanatory power as predictor variables. This result is consistent with recent findings [86] and social cognitive career theory [87], indicating that stronger positive results are associated with more frequent job information collection and career preparation behaviours. This offers empirical evidence that students exhibiting higher levels of goal setting, information collection, problem-solving, and future planning (i.e., the sub-factors of career preparation behaviour) engage in more career preparation behaviours.

College students who made greater practical efforts to achieve career goals engaged in more information collection behaviours. In addition, they were more actively involved in tool preparation and showed higher career decision-making self-efficacy. Accordingly, colleges should facilitate college students' independent career goal setting, active collection of job information, and instrumental career preparation behaviour to improve their career decision-making self-efficacy. Such guidance programmes should instigate college students' career preparation behaviour through motivational processes that can improve goal selection, job information collection, and problem-solving efficacy.

Third, in terms of the human development process, the college study period is optimal for making more realistic and concrete career decisions, and determining career possibilities by considering critical work conditions and one's desires and abilities [88]. Difficulties

related to career decisions encountered during this period can delay or significantly reduce career preparation behaviour. Anxiety has a particularly strong effect on college students' career development [89], with personality issues such as low self-esteem and identity conflicts exerting similarly powerful effects [82]. Furthermore, in unpredictable social environments, as with the rapid changes and instability of the occupational world, college students may experience various psychological difficulties when deciding on their career. Our study demonstrated that struggles with self-concept and identity negatively predict career decision-making self-efficacy; consistent with Kim (2005) [82].

Those who cannot decide on a career tend to have low self-esteem [90] and experience identity confusion [91]. Our findings indicate that when individuals enter college without a firmly established self-concept and identity (due to the South Korean educational climate that focuses only on entrance exams), they tend to experience career decision difficulties during this period. Therefore, college career counselling programmes should deliver content that will enable undergraduate students to establish their self-concept, personal identity, and career identity; these programmes need to be included in college curricula. Many studies show that career decision self-efficacy and career preparation behavior are related to or mediate academic achievement [92–95]. Additionally, there are studies that show the relationship between career preparation behavior and academic achievement [96–100].

Therefore, in order to improve academic achievement, it is necessary to develop a curriculum and an education system suitable for the times, and which combine professionalism and practicality to strengthen career decision-making self-efficacy and career preparation behavior.

According to Freeman et al. (2017), career courses have a positive effect on participants' emotional state, motivated by focusing on the career decision-making self-efficacy process and career planning [101]. Here, career identity [101] is an essential factor affecting college students' academic performance and life satisfaction [102]; therefore, college career counselling programmes should foster this. Accordingly, curricula that incorporate college career counselling programmes should be developed. Recently, the Ministry of Education in Korea implemented the 'University Career Exploration Credit System' in relation to careers, and conducted a pilot roll-out in 10 colleges in 2020 [103]. As part of the career preparation action, this policy helps students prepare easily and without burden for career exploration activities during the semester. Therefore, such a system should be implemented at all universities, and each university should invent its own measures to enable students to continuously enhance their sense of self-efficacy when deciding on their career while attending school and preparing for their career.

Limitations and Suggestions for Future Research

While the present findings have many implications for career-related counselling, education, and research focusing on college students, this study has several limitations. First, this research may have generalisability issues due to the application of convenience sampling to recruit college students in some regions of Korea. Since other regional and academic characteristics may affect the results, follow-up studies should verify the present findings through nationally representative sampling.

Second, this study's findings may not be completely reliable because it measured significant variables using a self-reported online questionnaire. For example, individuals' perceptions of evaluation might have been affected by a social desirability bias. Therefore, a more reliable measurement method might be necessary to fully understand the effects of multidimensional psychological attributes. Moreover, questionnaire response errors and distortions are possible because participants' interest and concentration may have waned during the latter part of the questionnaire. As this was a strictly quantitative measurement of individuals' psychological attributes, future studies should also collect qualitative data, perhaps by conducting in-depth interviews with undergraduate students in college career counselling centres.

Third, this study was conducted with the purpose of identifying the independent variables with high predictive ratings that affect career decision-making self-efficacy by set-

ting career preparation behaviour and career decision difficulties as independent variables through SEM. Career preparation behaviour is the strongest predictor of career decision-making self-efficacy. Future research should incorporate a structural equation model that can evaluate the relationship between sub-factors of career preparation behaviour and career decision-making self-efficacy to reveal more specific influences.

## 5. Conclusions

This study investigated how career decision-making self-efficacy—reported as a crucial factor in career development—affects the relationship between career preparation behaviour and career decision difficulties, which are vital during college life. We identified connections between South Korean college students' career decision-making self-efficacy, career preparation behaviour, and career decision difficulties. Compared to American and Japanese college students, Korean college students struggle more when choosing a career path, and have a higher rate of failure when focusing on themselves during the career preparation process [12]. Career probation and career exploration probation for Korean college students are becoming important social issues; this is related to students' lack of self-understanding and career self-identity. Therefore, it is recommended to design a career counselling programme in universities to establish self-concept and identity.

Our discoveries provide fundamental data for universities to guide career preparation behaviour and construct career guidance systems. Our research results also have implications for career-related public service policies at the national level.

**Author Contributions:** Conceptualisation, S.L. (Sanghee Lee); methodology, J.J. and S.B.; validation, J.J. and S.L. (Songyi Lee); investigation, S.L. (Songyi Lee), resources, S.L. (Songyi Lee); data curation, J.J. and S.B.; writing—original draft preparation, S.L. (Sanghee Lee) and J.J.; writing—review and editing, S.L. (Sanghee Lee); visualisation, J.J. and S.B.; supervision, S.L. (Sanghee Lee). All authors have read and agreed to the published version of the manuscript.

**Funding:** This research received no external funding.

**Institutional Review Board Statement:** The study was conducted in accordance with the guidelines of the Declaration of Helsinki. We used an anonymous coding system that makes identification of subjects impossible. Therefore, the authorisation of the Institutional Review Board was not necessary.

**Informed Consent Statement:** The purpose of this study was clearly explained to the participants before proceeding with the survey questionnaire. This was followed by a statement to the effect that, by returning the survey questionnaire, they would be deemed to have provided their consent. The authors followed COPE.

**Data Availability Statement:** The authors can provide data upon request.

**Conflicts of Interest:** The authors declare no conflict of interest.

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
