# Peer review of "The Relationship between Career Decision-Making Self-Efficacy, Career Preparation Behaviour and Career Decision Difficulties among South Korean College Students"

_sustainability, doi:10.3390/su142114384_

Round 1

Reviewer 1 Report

Introduction:

line 36-38: needs bibliographic reference.

The absence of self-efficacy in career decision making, poor career preparation behaviour and career decision difficulties among university students is discussed, although no relevant data from previous studies on the consequences of this and the impact it has on students is provided.

Does the fact that the students are from South Korea add any uniqueness to the study? The characteristics of these students are not compared with those of other nationalities. This would be a necessary point in this section.

Methods: Participants: It is somewhat enriching that the study represents different academic years, different age ranges as well as different fields of knowledge. However, there seem to be important differences in the distribution. It is well known that these categories have a decisive influence on the research variables. Statistical analysis between the categories is suggested in order to observe the homogeneity of the analysed sample.

Procedure: The procedure was carried out online, but how did the authors ensure that the answers were given by the participants in question? Through which platform were the questionnaires sent to them? 

It is necessary to provide the numerical record and the institution or bioethics committee that approved the research.

Results and Discussion: The authors discuss their results with other previous studies, using recent literature. We suggest that the authors include aspects concerning the practical implications derived from these results. Currently, there are educational policies aimed at enhancing the transversal competences of degrees in Higher Education. How could the results achieved be used in the task of the teacher?

Conclusion: To be rewritten. They do not respond faithfully to the objectives set out in the study. Aspects are included that have to do with practical implications and future lines of research, which should appear earlier in the manuscript.

Author Response

  1. line 36-38: needs bibliographic reference.

Response: We have revised our manuscript accordingly.

  1. The absence of self-efficacy in career decision making, poor career preparation behaviour and career decision difficulties among university students is discussed, although no relevant data from previous studies on the consequences of this and the impact it has on students is provided.

Response:  As you pointed out, we have updated the references (lines 56-62).

  1. Does the fact that the students are from South Korea add any uniqueness to the study? The characteristics of these students are not compared with those of other nationalities. This would be a necessary point in this section.

Response: We have revised our manuscript accordingly by adding Sohn (2018) and Jeong (2022) (lines 53-62, 102-104).

  1. Methods: Participants: It is somewhat enriching that the study represents different academic years, different age ranges as well as different fields of knowledge. However, there seem to be important differences in the distribution. It is well known that these categories have a decisive influence on the research variables. Statistical analysis between the categories is suggested in order to observe the homogeneity of the analysed sample.

Response: Following this point, we have conducted related analysis and mentioned the results of this performance in the preliminary and descriptive analysis in the Results section. Please refer to p.7.

  1. Procedure: The procedure was carried out online, but how did the authors ensure that the answers were given by the participants in question? Through which platform were the questionnaires sent to them? 

Response: We have revised the procedure section for greater clarity.

  1. It is necessary to provide the numerical record and the institution or bioethics committee that approved the research.

Response: We performed the numerical record. In terms of gaining IRB approval, we found this to be unnecessary because we used an anonymous coding system that makes the identification of subjects impossible. We conducted our study in compliance with the Committee on Publication Ethics (COPE) as well as the guidelines of the Declaration of Helsinki. We obtained signed informed consent forms from all participants after thoroughly explaining the study’s aims and procedures. Moreover, all researchers have received training and certification for performing ethical research involving human subjects. We can provide a translated copy and original copy of the certification upon request. (https://www.irb.or.kr/UserMenu01/Summary.aspx)

  1. Results and Discussion: The authors discuss their results with other previous studies, using recent literature. We suggest that the authors include aspects concerning the practical implications derived from these results. Currently, there are educational policies aimed at enhancing the transversal competences of degrees in Higher Education. How could the results achieved be used in the task of the teacher?

Response: Following the point made by the reviewer, we have added more practical implications of this study to the Discussion section. Please check lines 359–365 and 432–439.

  1. Conclusion: To be rewritten. They do not respond faithfully to the objectives set out in the study. Aspects are included that have to do with practical implications and future lines of research, which should appear earlier in the manuscript.

Response: We have revised our manuscript accordingly. Please check lines 53–62 and 473–479.

Reviewer 2 Report

Dear authors, 

Congratulations for your research about the relationship between career decision-making self-efficacy, career preparation behavior and career decision difficulties among South Korean college students.

I believe it offers an important contribution for research on career development and decision making, therefor I read your paper with curiosity. Along the reading, I have identified a set of questions and concerns that I believe should be addressed to increase the quality and transparency of your research. I wish you good luck.

Overall comments

Theoretical background

1.     Some papers / references / citations used in to theoretically support your paper are very dated. For example, in the section about Career Decision-Making Self-Efficacy only two references were published in the last 20 years. This is worrisome because it makes the reader question if the authors are ware of recent developments in the literature about the variables included in the research model. More recently published literature should be included.

2.     In the results section you presented, in detail, data about the sub-dimensions of the variables under study. However you do not sufficiently deepen these sub-dimensions in the theoretical background. Variables conceptualization needs to be more in-depth.

3.     You presented two research questions as a framework for the study. However, the arguments presented about the potential relationships between the variables justify, in my opinion, the presentation of hypotheses allowing to go beyond an exploratory approach.

Method

1.     The scales used to measure the variables of interest seem appropriate and have very good reliabilities, however it seems to be interesting the inclusion of CR and AVE’s estimates.

2.     Since you have collected data at a single point in time, your data may suffer from same source bias/ common method variance. I think you should elaborate on this and provide information about a single factor model when reporting the statistics for the measurement model. Regarding this, it is not clear for me if you have modelled the latent variables as second order construct or not (in the multidimensional variables).

3.     Please include information about Likert’s scales reference points.

Results

1.     I wonder if some of the participants’ characteristics (gender, age, …) are related to the variables of interest and could be controlled for in the analysis. Did you run Spearman correlation to check on this?

2.     Table 4 is not easy to read. I suggest you introduce the names of the variables in the first column as well, you need attempt to graphically harmonize the information in order to make the table more intelligible.

Discussion & Conclusions

1.     Your discussion and conclusion are aligned with the results obtained. The theoretical and practical implications also seem adequate. I suggest that you elaborate more on the limitations of your study.

Author Response

  1. Some papers / references / citations used in to theoretically support your paper are very dated. For example, in the section about Career Decision-Making Self-Efficacy only two references were published in the last 20 years. This is worrisome because it makes the reader question if the authors are ware of recent developments in the literature about the variables included in the research model. More recently published literature should be included.

Response: We have revised our manuscript accordingly.

  1. In the results section you presented, in detail, data about the sub-dimensions of the variables under study. However you do not sufficiently deepen these sub-dimensions in the theoretical background. Variables conceptualization needs to be more in-depth.

Response: As you pointed out, we have updated the variables conceptualization.

  1. You presented two research questions as a framework for the study. However, the arguments presented about the potential relationships between the variables justify, in my opinion, the presentation of hypotheses allowing to go beyond an exploratory approach.

Response: Generally, studies that establish and verify hypotheses are based on previous studies, such that hypotheses can be established with sufficient theoretical background. Other studies set research questions; this study was conducted by setting research questions because its scope is wide and it is not a direct verification study, rather based on a theory. We would appreciate it if you could consider this.

  1. The scales used to measure the variables of interest seem appropriate and have very good reliabilities, however it seems to be interesting the inclusion of CR and AVE’s estimates.

Response: CR and AVE’s estimates were mentioned in the measurement model of the Results section; please refer to p.7.

  1. Since you have collected data at a single point in time, your data may suffer from same source bias/ common method variance. I think you should elaborate on this and provide information about a single factor model when reporting the statistics for the measurement model. Regarding this, it is not clear for me if you have modelled the latent variables as second order construct or not (in the multidimensional variables).

Response: As the reviewer pointed out, we have provided the related information; please refer to p. 6 and Figure 1.

  1. Please include information about Likert’s scales reference points.

Response: We have revised our manuscript accordingly; please refer to p. 4 and p. 5.

  1. I wonder if some of the participants’ characteristics (gender, age, …) are related to the variables of interest and could be controlled for in the analysis. Did you run Spearman correlation to check on this? Response: We have added the related information to the preliminary and descriptive analysis in the Results section. Please refer to p. 7.
  2. Table 4 is not easy to read. I suggest you introduce the names of the variables in the first column as well, you need attempt to graphically harmonize the information in order to make the table more intelligible.

Response: We have revised our manuscript accordingly; please refer to p. 9.

  1. Your discussion and conclusion are aligned with the results obtained. The theoretical and practical implications also seem adequate. I suggest that you elaborate more on the limitations of your study.

Response: The paper has been updated regarding the limitations (lines 462–467).

Round 2

Reviewer 1 Report

The authors have done a good job in improving the previous version of the manuscript, although there are some procedural aspects to be clarified.

It is striking that in the midst of the covid 19 pandemic, they were able to recruit a sample size with so few losses.

It would be advisable to detail the procedure followed for the questionnaire to be completed by so many participants in a single week in this social context.

In addition, please detail whether the questionnaire was validated and adapted by experts.

Finally, it would be useful to discuss the results with other studies that indicate the effects of these variables on academic performance.

Author Response

  1. It is striking that in the midst of the covid 19 pandemic, they were able to recruit a sample size with so few losses.

It would be advisable to detail the procedure followed for the questionnaire to be completed by so many participants in a single week in this social context.

Response: Thank you for your feedback. We have added additional detailed procedures for better clarification. We actually posted a link to the questionnaire on the e-class notice board for a total of 658 students from two colleges to easily have access, and 350 students responded. From this number, we eliminated 9 responses (which were unanswered/error) to reach our final sample size. Please refer to p.6.

  1. In addition, please detail whether the questionnaire was validated and adapted by experts.

Response: We have used a validated questionnaire and clarified this in the method section. Please refer to p.4~5.

  1. Finally, it would be useful to discuss the results with other studies that indicate the effects of these variables on academic performance.

Response: We have revised accordingly. Please refer to p. 12

Reviewer 2 Report

I consider that the questions/problems I raised have been properly answered or resolved.

Author Response

I consider that the questions/problems I raised have been properly answered or resolved.

Response: Thank you so much.